# Inflammatory Control of Viral Infection

**DOI:** 10.3390/v15071579

**Published:** 2023-07-20

**Authors:** Sukanya Chakravarty, Ritu Chakravarti, Saurabh Chattopadhyay

**Affiliations:** 1Medical Microbiology and Immunology, University of Toledo College of Medicine and Life Sciences, Toledo, OH 43614, USA; sukanya.chakravarty@rockets.utoledo.edu; 2Physiology and Pharmacology, University of Toledo College of Medicine and Life Sciences, Toledo, OH 43614, USA; ritu.chakravarti@utoledo.edu

**Keywords:** IRF3, NF-κB, inflammation, innate immunity, interferon, antiviral, RIKA

## Abstract

Inflammatory responses during virus infection differentially impact the host. Managing inflammatory responses is essential in controlling viral infection and related diseases. Recently, we identified a cellular anti-inflammatory mechanism, RIKA (Repression of IRF3-mediated inhibition of NF-κB activity), which controls viral inflammation and pathogenesis. The RIKA function of IRF3 may be explored further in other inflammatory diseases beyond viral infection.

Viral infection rapidly turns on a plethora of genes, many of which are critical for successful clearance of the virus. Interferons (IFNs) and IFN-stimulated genes (ISGs) are the most prominent ones that inhibit a wide range of viruses [1,2]. Virus-infected cells, in addition to antiviral genes, induce inflammatory genes, which are critical for innate immune responses. Deficiency or imbalance in antiviral or inflammatory arms of immune responses causes undesired outcomes, i.e., viral diseases. Cellular regulators are in place to control both sides of innate immune responses; carefully managing both arms can help avoid these undesired outcomes.

Viral components are detected in infected cells by pattern recognition receptors (PRRs), e.g., TLRs, RLRs, cGAS, etc. [3,4]. Upon binding viral nucleic acids, PRRs trigger downstream signaling pathways to activate IRF3 and NF-κB, transcription factors that cooperatively act to induce type-I as well as type-III IFNs. PRR-activated IRF3 causes expression of antiviral genes, whereas NF-κB activation leads to pro-inflammatory gene expression [5,6,7]. In the early phase, inflammatory genes help protect against infection; however, in the later stage, they can cause cell and tissue damage by inducing cytokine storm, cell death, etc. [8,9]. Inflammation has beneficial effects on tissue homeostasis and resolution of infection; however, harmful effects are not completely clear. It is largely appreciated that viral load is the predominant factor contributing to viral diseases. However, it is becoming increasingly clear, particularly from studies on SARS-CoV-2 and influenza A virus (IAV) infection, that virus-induced lung inflammation contributes significantly to viral pathogenesis [10,11,12,13]. Neuroinflammation by herpesviruses and flaviviruses, as well as cardiac inflammation by reoviruses, contribute to lethal viral diseases [14,15,16,17,18,19,20]. Controlling virus-induced inflammation when it is harmful to the host can suppress viral pathogenesis.

IRF3 is antiviral against a wide range of viruses; IRF3 deficiency causes increased viral replication and susceptibility to viral infection [21,22]. IRF3 activation causes the transcriptional induction of IFNβ, a cytokine, which is secreted from infected cells and acts by autocrine and paracrine signaling to trigger a transcriptional induction of ISGs [23,24]. For transcriptional activity, IRF3 is phosphorylated by TBK1, allowing its translocation into the nucleus. In addition to the transcriptional response, virus-activated IRF3 causes apoptotic cell death in a transcription-independent pathway, RLR-induced IRF3-mediated Pathway of Apoptosis (RIPA, Figure 1) [25,26,27,28]. For RIPA, IRF3 undergoes linear ubiquitination by LUBAC, leading to its interaction with pro-apoptotic protein BAX, causing translocation into mitochondria. An IRF3 mutant, defective in transcriptional response but active in RIPA, can inhibit viral replication in cells and mice.

Recently, we uncovered a new function of IRF3, Repression of IRF3-mediated NF-κB Activation (RIKA), inhibiting NF-κB activity to suppress inflammatory gene expression (Figure 1) [29]. IRF3 knockout cells express elevated NF-κB-dependent genes, e.g., the inflammatory target genes, upon Sendai virus (SeV), influenza A virus (IAV), and murine hepatitis virus (MHV) infection compared to the control cells. Microarray analyses using control and IRF3-knockdown cells further validated these results. In addition to virus infection, we used non-viral stimuli, e.g., polyI:C, LPS, and cGAMP, to demonstrate a RIKA-mediated suppression of inflammatory genes. Mechanistically, IRF3 interacts with the NF-κB-p65 subunit directly, sequestering in cytosol to inhibit its translocation into the nucleus. Using deletion analyses, we mapped the domains responsible for the IRF3:NF-κB-p65 interaction. IRF3 mutants, defective in either transcriptional activity (IRF3-S1), or RIPA, or both (IRF3-M1), were able to interact with NF-κB-p65 and exhibit RIKA (Figure 2). Surprisingly, IRF3-M1, like IRF3-Wt, inhibits viral replication, demonstrating the antiviral activity of RIKA. Finally, IRF3 knockout mice, infected intranasally with SeV, exhibit enhanced inflammatory genes in the infected lungs compared to Wt mice. Therefore, virus-induced inflammation is a critical determinant of viral pathogenesis, and the RIKA branch of IRF3 contributes to the optimal antiviral activity of IRF3.

Although discovered in the context of virus infection, RIKA has implications beyond viral diseases. Many bacteria rely on cellular inflammatory responses, and RIKA may promote their replication and pathogenesis. RIKA plays a key role in alleviating high-fat diet (HFD)-induced liver injury [30]. In the HFD-induced liver injury model, IRF3 interacts with NF-κB-p65, inhibiting its activation and inflammatory gene expression. Knock-in mice, expressing the IRF3-S1 mutant, transcriptionally inactive but active in RIKA, suppress HFD-induced liver injury. A RIKA-like activity of IRF3 controls HFD-induced hepatic steatosis and insulin resistance, in which IRF3 interacts with IKKβ to suppress hepatic inflammatory responses [31]. IRF3 promotes LPS-induced septic shock in mice; IRF3 knockout mice are resistant to LPS-mediated sepsis compared to Wt mice [32,33]. However, the molecular mechanisms are unclear; whether RIKA-like activities play any role in the sepsis-promoting function of IRF3 needs to be investigated. Many cancer cells rely on NF-κB activity for survival since NF-κB is involved in anti-apoptotic gene expression [34,35,36]. The RIKA function of IRF3 may be explored for inhibiting NF-κB activity to promote cancer cell death. Future studies involving pathway-specific IRF3 mutants (as shown in Figure 2) can be utilized to elucidate the role of RIKA in diseases beyond viral infection. IRF3-derived small peptides, designed to specifically bind NF-κB, may have anti-inflammatory activities, with therapeutic potential against inflammatory diseases.

A major question that remains to be answered is whether these pathways of IRF3 operate simultaneously or sequentially in the infected cells. It is speculated that in the early stage, cells would primarily rely on the transcriptional pathway of IRF3 to mount antiviral responses, which also require NF-κB activity. In the second phase, RIKA may be critical to curb the undesired inflammatory responses. Later in the infection, as we have also shown in SeV-infected cells, RIPA-mediated apoptotic killing can help eliminate the infected cells. It remains to be seen whether these pathways are operational in all cells or if there is cell type-specific activation of IRF3 functions. The pathway-specific mutants will be critical tools to tease them apart in future studies. Finally, our investigation can be expanded to other IRFs and viral IRFs to examine the specificity of IRF-dependent cellular anti-inflammatory responses. In summary, the identification of RIKA opens new avenues for addressing various critical questions related to IRF biology.

## Figures and Tables

**Figure 1 viruses-15-01579-f001:**
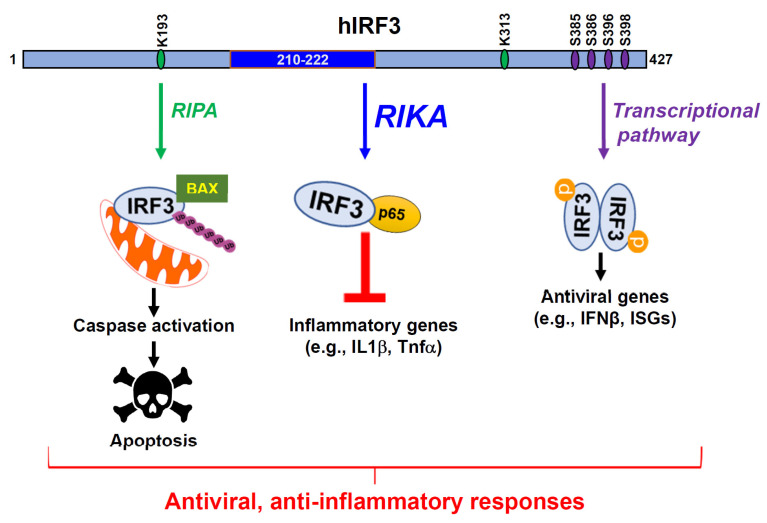
RIKA, a new anti-inflammatory function of IRF3. IRF3 is widely known for its transcriptional function involving the production of IFNβ and ISGs which antagonize various stages of viral replication. A non-transcriptional function of IRF3, RIPA (RIG-I-like receptor-induced IRF3 mediated Pathway of Apoptosis), was uncovered through our studies, involving IRF3 apoptotically killing virus-infected cells. In RIPA, linear ubiquitination of IRF3 upon virus infection results in its activation and translocation into the mitochondria. Subsequent activation of downstream caspases results in apoptotic cell death. RIKA, a new non-transcriptional function of IRF3, downregulates inflammatory gene expression. IRF3 domain, comprising 210–222 amino acids, interacts with the p65 subunit of NF-κB to sequester it in the cytosol. As a result, induction of inflammatory cytokines such as IL1β, Tnfα is restricted, thus suppressing the inflammatory responses and pathogenesis caused by the overproduction of these cytokines.

**Figure 2 viruses-15-01579-f002:**
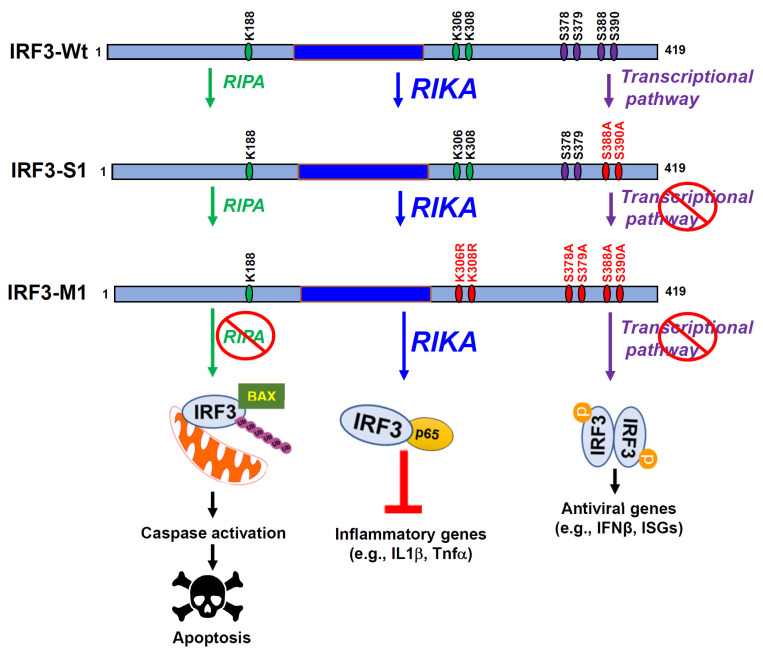
Pathway-specific mutants of IRF3 and their functions. Mouse IRF3 (IRF3-Wt) with its critical amino acids and pathway-specific mutants (IRF3-S1 and IRF3-M1) are shown. Wt IRF3 is capable of transcriptional, RIPA, and RIKA functions, IRF3-S1 is active in RIPA and RIKA but inactive in transcriptional activity, and IRF3-M1 is active only in RIKA but inactive in transcriptional and RIPA activities.

## Data Availability

Not applicable.

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
