# Peer review of "Inflammatory Control of Viral Infection"

_viruses, 2023, doi:10.3390/v15071579_

Round 1

Reviewer 1 Report

Chakravarty et al’s commentary about the inflammatory control of viral infection was overall very clear and highlights a novel function of IRF3. Overall, the commentary was good but requires a few minor modifications:

1.     The following sentences are redundant and should be combined: “IRF3 knockout cells express elevated NF-kB-dependent genes, e.g., the inflammatory target genes, upon virus infection, compared to the wild-type (Wt) cells. Using Sendai virus (SeV), influenza A virus (IAV), and murine hepatitis virus (MHV), we demonstrated that IRF3-deficient cells express enhanced inflammatory genes upon virus infection.

2.     The following sentences are redundant and should be combined.” IRF3 mutants defective in either transcriptional activity or RIPA were able to interact with NF-kB-p65 and exhibit RIKA. IRF3 mutant (M1), defective in both transcriptional and RIPA activities, interacts with NF-kB-p65, and when expressed in IRF3-null cells, exhibits RIKA-mediated suppression of inflammatory genes.”

3.     IRF3 also induces type III IFNs and not just type I IFNs, which would be key in respiratory infections.

4.     In the text two IRF3 mutants are described (IRF3 mutant (M1) and IRF3-S1 mutant). These mutants should be shown in the figure to make it clearer what they are.

5.     It would be nice to have more details about RIKA and its timing. Does it occur before interferon/ISG induction? Is it in parallel? Does it only happen later in the response? This was not clear from the text and would be helpful to include. A figure showing how this occurs could also be helpful.

6.     The concluding paragraph is a bit confusing. This should be re-written to make it clearer where RIKA plays a role outside of viral infection. Additionally, the text ends very abruptly and it would be nice to have a few concluding sentences.

1.     Overall the text is clear however, there are a few grammar errors and the text should be re-read carefully to correct for these.

Reviewer 2 Report

It is nicely written. Some statements need references for supporting, for example, the statement "Many cancer cells rely on NF-kappaB activity for survival since NF-kappaB is involved in anti-apoptotic gene expression." needs more citations.
